# Query Augmentation with Brain Signals

## ABSTRACT

In the information retrieval scenario, query augmentation is an essential technique to refine semantically imprecise queries to align closely with users' actual information needs. Traditional methods typically rely on extracting signals from user interactions such as browsing or clicking behaviors to augment the queries, which may not accurately reflect the actual user intent due to inherent noise and the dependency on initial user interactions. To overcome these limitations, we introduce **Brain-Aug**, a novel approach that decodes semantic information directly from brain signals of users to augment query representation. Brain-Aug explores three-fold techniques: (1) Structurally, an adapter network is utilized to project brain signals into the embedding space of a language model, allowing query augmentation conditioned on both the users' initial query and their brain signals. (2) During training, we use a next token prediction task for query augmentation and adopt prompt tuning to efficiently train the brain adapter. (3) At the inference stage, a ranking-oriented decoding strategy is implemented, enabling Brain-Aug to generate augmentations that improve ranking performance. We evaluate our approach on multiple functional magnetic resonance imaging (fMRI) datasets, demonstrating that Brain-Aug not only produces semantically richer queries but also significantly improves document ranking accuracy, particularly for ambiguous queries. These results validate the effectiveness of our proposed Brain-Aug approach, and reveal the great potential of leveraging internal cognitive states to understand and augment text-based queries.

## CCS CONCEPTS

• **Human-centered computing** → **HCI design and evaluation methods**; • **Computing methodologies** → **Artificial intelligence**; • **Information systems** → **Users and interactive retrieval**.

## KEYWORDS

Query augmentation, Prompt tunning, Brain-computer interface (BCI)

## 1 INTRODUCTION

Understanding users' intentions is the key to effective search engines. In the interactions between users and search engines, queries play an important role in presenting the users' intentions and for search engines to retrieve relevant documents. However, search engine users often struggle to express their information needs precisely, resulting in queries that are short [21], vague [53], or inaccurately phrased [11], which compromise the retrieval effectiveness.

*ACM MM, 2024, Melbourne, Australia*
© 2024 Copyright held by the owner/author(s). Publication rights licensed to ACM.
ACM ISBN 978-x-xxxx-xxxx-x/YY/MM
https://doi.org/10.1145/nnnnnnn.nnnnnnn

To address this problem, query augmentation emerges as a crucial technique to refine the original queries into more effective expressions [23, 33]. Traditionally, this reformulation process relies heavily on external document information such as expanding the query with contents from documents users have engaged with [2, 9, 43].

The advent of neurophysiological interfaces offers a novel source of data to understand users' search intentions [36, 57]. In information retrieval (IR) scenarios, several studies have revealed that brain signals can be used to predict users' relevance perception [14, 44, 58] and cognitive state [38]. These advances open new avenues in using brain signals as an alternative to conventional signals for query augmentation. Existing studies have investigated the use of brain signals to predict the relevance of perceived input [13], which can be further used to extract relevant content for query augmentation [55, 56]. However, the existing process of query augmentation still relies on the quality of initially retrieved documents and cannot kick off before potentially unsatisfactory interactions with initial documents.

In this paper, we introduce **Brain-Aug**, a novel approach to query augmentation. It leverages brain signals to directly refine user-submitted queries by decoding semantic information embedded in neural activity. Brain-Aug incorporates three core elements: (i) Model Architecture: Brain-Aug employs a mapping network to transform brain signals into the input space of transformer model. This allows the model to generate query conditional on both the brain signals and the initial query simultaneously, effectively integrating neural data with computational language processing. (ii) Training Protocol: We develop a specialized pre-training alignment task tailored for brain signals and a fine-tuning process specifically for query augmentation. This dual training strategy enhances the model's ability to decode users' intentions from their neural signals. (iii) Ranking-Oriented Inference: During inference, Brain-Aug implements a ranking-oriented decoding strategy that utilizes inverse document frequency (IDF) to generate query continuations. This method ensures that the augmented words not only fit the context but also possess distinctive characteristics to improve ranking performance.

We conduct comprehensive experiments to validate the effectiveness of Brain-Aug. Using a variety of functional magnetic resonance imaging (fMRI) datasets and different retrieval systems, our results robustly demonstrate that Brain-Aug can accurately interpret user intentions and enhance search engine performance. Our approach not only significantly outperforms traditional query augmentation methods but also enhances their efficacy when combined with these methods. Furthermore, we performed both quantitative and qualitative analyses to deeply analyze Brain-Aug's capabilities. Our analysis reveals that Brain-Aug can effectively augment ambiguous queries, creating clearer and more precise queries that significantly boost retrieval effectiveness.

In summary, our contributions are as follows: (1) We introduce Brain-Aug, a novel architecture that enhances query representation by incorporating brain signals as an additional input. We have devised training and inference protocols aimed at refining queries with

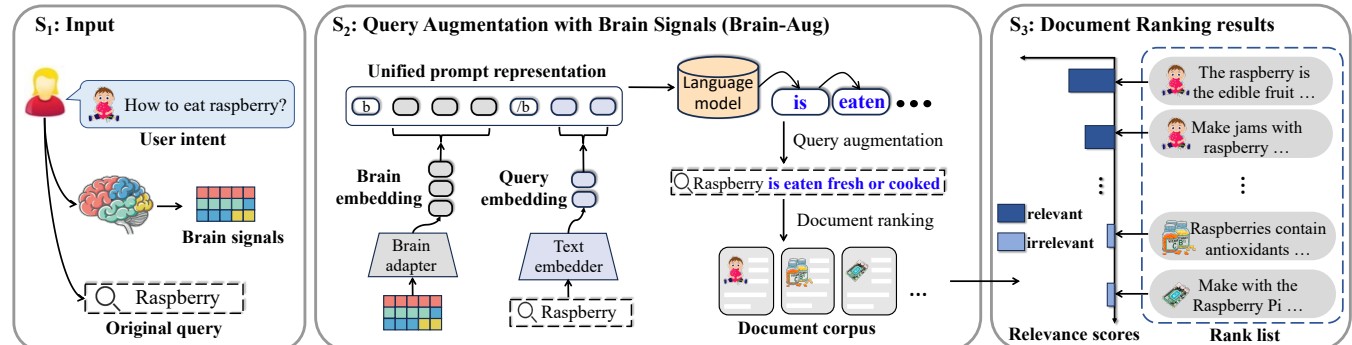

**Figure 1: The procedure of query augmentation with brain signals (Brain-Aug). Brain-Aug constructs a unified prompt representation that jointly models the brain responses and original queries. With the unified prompt representation as input, a language model is adopted to generate the continuation of the original query for its augmentation.**

greater semantic precision and enhanced discriminative capability across different documents. (2) We demonstrate the effectiveness of Brain-Aug, showing that Brain-Aug refines queries to align more semantically closely with the search intent. We further show that the augmented query can be used to improve search performance in terms of document ranking. (3) We analyze the performance gain achieved by Brain-Aug against its controls and unsupervised baselines. We observe that Brain-Aug is more effective in cases where the original query is probed to be ambiguous.

## 2 RELATED WORK

### 2.1 Query augmentation

Traditionally, query augmentation can be categorized into two types: based on pseudo-relevance signals [7, 23] and based on user signals [26]. Approaches based on pseudo-relevance signals usually treat top-ranked documents in the initial retrieval step as relevant. Based on these relevant documents, Rocchio Jr [48] and Lavrenko and Croft [23] adopt a vector space model and a language model for refining the query representation to be closer to the top-ranked documents, respectively. In contrast, approaches based on user signals usually integrate information from documents the user has previously interacted with or queries they submitted historically. E.g., Chen et al. [9] and Ahmad et al. [2] build a sequence model to extract semantic representations from historical clicked documents to refine the query representation. Existing methods, either based on pseudo signals or user signals, are limited by their reliance on the quality of the documents and the accuracy of estimating their relevance.

### 2.2 Neuroscience and IR

There is increasing literature that adopts neuroscientific methods into IR scenarios [10, 16, 30, 39, 45]. For example, Chen et al. [10] built a prototype in which users can interact with the search systems with a brain-computer interface. Allegretti et al. [3], Michalkova et al. [36], Moshfeghi et al. [38] conducts a series of work to study the cognitive mechanisms involved in the process of information retrieval. A common finding observed by existing literature is that[3, 14] brain signals can be utilized to as a relevance indicator. This indicator can be employed for query rewriting [13, 56]. Although this paradigm

has been shown to be effective, it still relies on the quality of the retrieved documents. On the other hand, other studies have demonstrated that semantics could be decoded to some extent with brain signals such as fMRI [54, 61] and magnetoencephalogram (MEG) [12]. However, there is currently a lack of research investigating the utilization of the decoded semantics for query augmentation.

## 3 METHOD

We first formalize the query augmentation task and then present Brain-Aug, including its architecture, training objective, and inference process.

### 3.1 Task formalization

In search engines, users' input queries are often unclear, failing to accurately reflect their true intentions. As brain-computer interface techniques become increasingly cost-effective and wearable, this paper explores the potential of leveraging brain signals to enhance the queries written by users. By incorporating the brain neural data, we aim to capture and reflect user intentions more precisely, augment queries, and thereby improve the accuracy of search results.

The *input* to the task of augmenting queries with brain signals is a query submitted by a user and the brain signals associated with the query context. $Q$ is used to denote the query composed of $n$ tokens: $Q = \{q_1, q_2, \ldots, q_n\}$. $B = \{b_1, \ldots, b_t\} \in \mathbb{R}^{t \times c}$ represent the brain signal, which is a sequence of features extracted from fMRI data, where $c$ is the number of fMRI features and $t$ is the number of time frames that brain recordings are collected.

Given the input query and brain signals, the *task* is to learn an autoregressive function $F$ to refine the query based on the user's cognitive process. $F$ generates a query continuation $M = \{m_1, \ldots, m_k\}$, which will be concatenated to the initial query $Q$ as an augmentated query. Let $m_i$ be the $i$-th token of $M$, the generation process is formalized as:

$$m_i = F(\{q_1, \ldots, q_n, m_1, \ldots, m_{i-1}\}, B; \Theta), \quad (1)$$

where $\Theta$ is the model parameters of $F$.

The effectiveness of query augmentation is measured *extrinsically* using the document ranking performance. Formally, let $\mathcal{D}$ be a document corpus and $G$ be a ranking model (e.g., BM25 [47],

RepLLaMA [32]). The ranking model $G$ estimates a ranking score $G(\{Q, M\}, d)$ for each document $d \in \mathcal{D}$ and the document ranking performance can be measured by a ranking-based metric such as normalized discounted cumulative gain (NDCG) [19] or mean average precision (MAP) [20].

## 3.2 Overall procedure

Fig. 1 provides an overview of the three-stage process of Brain-Aug: $S_1$ : Input to Brain-Aug consists of the original query and brain signals associated with the user's cognitive response within the query context. $S_2$ : Then a brain adapter is trained to align the representations of brain signals with the representation space of text embedding in the language model. This allows for creating a unified prompt representation that jointly models the brain responses and original queries. With the unified prompt representation as input, a language model is adopted to generate the continuation of the original query. A ranking-oriented inference method is utilized to enhance the generation process to improve the ranking performance. $S_3$ : In this case, the original query "Raspberry" (sampled from Pereira's dataset in our experiment) is augmented to "Raspberry is eaten fresh or cooked". Consequently, documents with a focus on the subtopic of "eating raspberry" are ranked higher than those on "raspberry's nutrition" or "raspberry Pi".

## 3.3 Model Architecture

Brain-Aug integrates the textual query with cognitive information derived from brain signals and inputs them into a Transformer model. The Transformer is used to generate query augmentations based on the context of initial queries and user brain signals. In the following, we describe how to map the brain signals and input it to Transformer.

First, since the brain signal extracted by fMRI cannot be directly processed by a pretrained language model, we devise a brain adapter $f_b$ to embed each brain representation $b_i \in B$ into the same latent space $\mathbb{R}^d$, which can be formulated as $v_i^B = f_b(b_i)$. We implement it as a neural network $f_b$ comprising (i) a MLP network $f_m$ with ReLU [15] as the activation function, and (ii) a position embedding $P = \{p_1, \ldots, p_t\} \in \mathbb{R}^{t \times c}$. Element-wise addition is applied where each position embedding $p_i \in P$ is added to its corresponding fMRI features $b_i \in B$. The multi-layer perceptron network $f_m$ is constructed with an input layer and two hidden layers. Formally, the fMRI features $b_i$ is mapped as:

$$v_i^B = f_b(b_i) = f_{mlp}(p_i + b_i). \tag{2}$$

where $i$ denotes the $i$-th time frame.

Then, we acquire the embeddings of the initial query. We feed the query's text $Q$ to the language model's embedding layer $f_q$ to transform the tokens into latent vectors $V^Q = \{v_1^q, \ldots, v_i^q, \ldots, v_n^q\} \in \mathbb{R}^{n \times d}$, where $n$ is the number of tokens, $d$ is the embedding size of the language model.

Finally, the brain embedding $V^B$ and the query embedding $V^Q$ are concatenated with embeddings of two special tokens, i.e., $\langle b \rangle$ and $\langle /b \rangle$, marking the beginning and end of the brain embedding, respectively. The two special tokens are randomly initialized as one-dimensional vectors aligned with the dimensional structure of token embeddings in the language model. As a result, the prompt sequence

$S$ can be represented as:

$$S = \{\langle b \rangle, v_1^B, \ldots, v_t^B, \langle /b \rangle, v_1^Q, \ldots, v_n^Q\}. \tag{3}$$

This sequence, integrating both brain information and textual data, can be input to the language model for generating the query continuation.

## 3.4 Training

To effectively leverage brain signals for query augmentation, we design a two-stage training process. The first is a unsupervised training stage and is to warm-up the brain adapter for aligning the brain input to the latent space of the language model. The second is a supervised learning stage and is to guide the model to decode semantic information from brain signals for query augmentation.

*3.4.1 Unsupervised training to warm-up the brain adapter.* We design an unsupervised warm-up stage to align the distribution of the brain embedding with that of the text token's embeddings, ensuring that the brain embedding is suitable as the input of a language model. We construct training pairs in an unsupervised manner. Each pair consists of a series of brain signals and the associated text. Formally, let $V^B$ be the mapped brain signals. Each $v_i^B \in V^B$ is trained to be close to the mean value of the corresponding query embeddings, i.e., $\frac{1}{n} \sum_{j=1}^{n} v_j^Q$. Mean square loss (MSE) loss is adopted for training, which can be formulated as:

$$L_{MSE} = \frac{1}{t} \sum_{i=1}^{t} \left( v_i^B - \frac{1}{n} \sum_{j=1}^{n} v_j^Q \right)^2. \tag{4}$$

The inclusion of a warm-up stage in the adapter training phase of a language model with multimodal input is crucial, as illustrated by Liu et al. [28]. Our experimental results corroborate these findings. We observe that omitting the adapter training phase can result in problems like unstable training, exemplified by gradient explosion, and inferior performance compared to models that include a warm-up stage.

*3.4.2 Query augmentation as next token prediction.* Given the input $S$ as formulated in Eq. (3), we train the model with the next token prediction task using a prompt tunning setup. Let $M^* = \{m_1^*, \ldots, m_k^*\}$ be the ground truth outputs. The language model is trained to predict $M$ on the condition of $S$. The training objective is to maximize the likelihood of generating the ground truth, which can be formulated as:

$$\max_{\Theta} = \sum_{i=1}^{k} \log(P_{LM}(m_i^* \mid \{m_1^*, \ldots, m_{i-1}^*\}, S; \Theta)), \tag{5}$$

where $\Theta$ is the model parameters. Constructing ground-truth labels presents a significant challenge, as our task assumes that users are not good at writing clear queries. Consequently, we cannot simply ask users to write a clearer ground-truth query. To address this issue, we hypothesize that an accurate representation of user intent corresponds to the documents they consider relevant. Therefore, we set the ground truth to be these relevant documents and train the model to reconstruct relevant documents based on user's brain signals and initial query. This approach effectively avoids the difficulty of having users directly annotate clear queries, as it is comparatively easier for them to identify relevant documents.

The training process follows the "prompt tuning" approach [29] by freezing the parameters of the language model and fine-tuning only the prompt representation $S$. This indicates that only the parameter of the brain adapter ($\Theta^{fb}$) and the parameter of the special tokens ($\Theta^{sp}$) are updated. In this way, we can train Brain-Aug efficiently with limited training data constructed from brain imaging datasets.

## 3.5 Ranking-oriented inference

During the inference stage, the generated continuations should also be able to distinguish between different documents. Therefore, we incorporate the IDF information [46] of each token in the vocabulary when generating query continuation $\hat{M} = \{\hat{m}_1, \ldots, \hat{m}_k\}$. Let $\text{IDF}(\hat{m})$ be the IDF of token $\hat{m}$, then the generation likelihood of each token in $\hat{m}_i \in \hat{M}$ during the inference stage can be estimated as:

$$P_{\text{inf}}(\hat{m}_i) = \frac{P_{\text{LM}}(\hat{m}_i) + \alpha \, \text{IDF}(\hat{m}_i)}{\sum_{m \in \text{Vocab}} (P_{\text{LM}}(m) + \alpha \, \text{IDF}(m))}, \qquad (6)$$

where $P_{LM}(m) = P_{LM}(m \mid \{\hat{m}_1, \ldots, \hat{m}_{i-1}\}, S; \Theta)$ represents the estimated likelihood of the next token $m$ given the previously generated tokens $\{\hat{m}_1, \ldots, \hat{m}_{i-1}\}$, $\alpha$ is a hyperparameter, Vocab indicates the language model's vocabulary. This approach ensures that the query's continuation is not only contextually relevant but also effective in distinguishing documents in the retrieval process.

# 4 EXPERIMENTAL SETUP

Next, we detail our experimental settings, which are designed to address three research questions: **(RQ1)** Is it possible to generate an augmented query with user's brain signals? **(RQ2)** Can we improve document ranking performance using the augmented query? **(RQ3)** How do brain signals improve different queries for document ranking? Together, these questions help us to understand the effectiveness of Brain-Aug to refine a query and improve ranking performance. Below, we describe the datasets and baselines. More implementation details are provided in Section A.4.

## 4.1 Datasets

Three publicly available fMRI datasets are adopted, namely Pereira's dataset [42], Huth's dataset [24], and the Narratives dataset [41]. We process the text stimuli in these datasets to transform them into ranking datasets consists of a document corpus and a set of queries. The dataset information is provided in Section A.1.

## 4.2 Data processing

We extract queries and documents from existing fMRI datasets following Izacard et al. [18] and Lee et al. [25]. Specifically, we select a text span in the document as a pseudo query and the corresponding document is treated as relevant for this query. Formally, for a document $D = \{w_1, \ldots, w_m\}$, we extract a span $Q = \{w_l, w_{l+1}, \ldots, w_r\}$ to form a relevant query-document pair $\{Q, D \backslash Q\}$, where $D \backslash Q = \{w_1, \ldots, w_{l-1}, w_{r+1}, \ldots, w_m\}$.

In Pereira's dataset, each document consists of 3-4 sentences, which are presented to the user as visual stimuli one by one. Due to the length of a sentence being too long as a query, we truncate the first one-third and two-thirds of the sentence to construct two queries for each sentence, resulting in 6-8 relevant query-document pair

for each document. In Huth's and Narratives datasets, continuous contents are presented to the user as auditory stimuli. We utilize a fixed time interval of 20 seconds, which corresponds to 10 fMRI scans, to segment the stimuli into documents. Then, smaller time intervals of 2, 4, and 6 seconds are employed to segment queries of varying lengths from the document. We provide more details and statistical data for the document corpus and queries constructed in each dataset in Section A.2.

Due to the variability in brain data across participants, we trained separate models for each participant and evaluated Brain-Aug using a five-fold cross-validation on each participant's data. The data samples are randomly split into five folds according to which document they belong to. Each fold of the cross-validation involves selecting one fold of the data as the test set, while the remaining four folds are split into training and validation sets. The sizes of the training, validation, and testing sets were roughly proportional to 3:1:1, respectively.

## 4.3 Training and evaluation setup

We train Brain-Aug with a next token prediction task. A data sample during this task consists of the query, its ground truth continuation, and corresponding brain signals. The ground truth continuation is selected as the textual content presented within a fixed period of time after the query (see Section A.2 for details). Taking into account the delayed effect of fMRI signals[37], we collect user's brain signals in a period of several seconds after the user perceives the textual content of the query. During this period, the user's brain representation has the potential to encode semantic information related to the query itself, as well as its continuation.

We first conduct *query generation analysis* to investigate the ability of Brain-Aug to generate query continuation that matches the ground truth label. The logarithm perplexity [34] is used to measure the likelihood of generating the ground truth continuation. The lower perplexity indicates the language model deems the ground truth continuation as more expected. We also investigate language similarity to demonstrate the extent to which the generated continuation is similar to the ground truth using the Rouge score [27].

Next, we augment the original query with its generated continuation and evaluate its performance in terms of *document ranking*. We employ document ranking metrics, including NDCG at different cutoffs (10 and 20) [19], Recall@20, and MAP [20].

## 4.4 Baselines and controls

Given the augmented query, we select two ranking models for document ranking, i.e., a sparse ranking model, **BM25** [47], and a dense ranking model, **RepLLaMA** [32]. To assess whether Brain-Aug helps document ranking, we compare its document ranking performance with several *baselines* and *controls*.

As *baselines* we select (i) **the original query**, and (ii) the query augmented with pseudo-relevance signals (denoted as **Unsup-Aug**). When using BM25 as the ranking model, we implemented RM3 [23] as Unsup-Aug, which expands the query by selecting relevant terms from the top-ranked documents in the initial retrieval. When using RepLLaMA as the ranking model, we implement Rocchio [7] as Unsup-Aug, which refines the query vector to be closer to the

top-ranked documents. (iii) We also reported the additional results by first using Brain-Aug, followed by Unsup-Aug, denoted as **Unsup+Brain-Aug**.

As *controls* we select variants or ablations of Brain-Aug. The first control is Brain-Aug without any brain input (denoted as **w/o Brain**), and thus the query continuation is generated solely depending on the original query and the language model. The second control is Brain-Aug with randomly sampled brain input (denoted as **RS Brain**). RS Brain involves sampling brain input that does not correspond to the query but is randomly selected from the same dataset. The last control is Brain-Aug without ranking-oriented generation in which the generation likelihood of each token is estimated without the IDF weight (denoted as **w/o IDF**).

## 5 EXPERIMENTS AND RESULTS

We first analyze the performance of the generated query continuation by comparing it with the ground truth label. Then we investigate the document ranking performance with Brain-Aug and examine the relationship between query features and their ranking performance.

### 5.1 Query generation analysis

Next, we evaluate the performance of Brain-Aug according to the similarity of the generated continuation and the ground truth label of continuation. The query generation analysis results are presented in Table 1. From Table 1, we have the following observations.

(1) Brain-Aug exhibits lower perplexity and higher Rouge-L than its ablations without brain input (w/o Brain) and randomly sampled brain signals as input (RS Brain). This indicates that the semantic information decoded from brain signals can be integrated with a query to construct a more effective prompt for generating query continuation.

(2) The overall perplexity and Rouge-L on the Pereira dataset are lower and higher than on the other two datasets, respectively. This implies that the Pereira dataset, derived from Wikipedia data, exhibits superior performance in the task of query generation compared to the other two datasets, which are based on spoken stories.

(3) The RS Brain outperforms w/o Brain across three datasets. Although RS Brain uses brain signals that do not correspond to the current query context, the unified prompt can enable generating content that aligns with the common data distribution of language usage in the dataset (e.g., all stimuli in Pereira's dataset are Wikipedia-style). On other other hand, w/o Brain is equivalent to a standard language model that generates continuations soly based on the query text. This difference explains RS Brain's superior performance compared to the w/o Brain. However, in the discussion in Section 5.2, we will show that this performance improvement in query generation does not necessarily lead to an improvement in document ranking.

**Answer to RQ1.** The results show that queries augmented with semantics decoded from brain signals are more aligned with the content of the relevant document with the help of brain signals.

### 5.2 Document ranking performance

#### 5.2.1 *Overall performance.* Table 2 shows the document ranking performance with original queries, queries augmented with unsupervised signals (Unsup-Aug), and queries augmented with brain signals (Brain-Aug). We observe:

**Table 1: Query generation performance averaged across participants in different datasets. Best results in boldface. * indicates $p \leq 0.05$ for the paired t-test of *Brain-Aug* (Ours) and the controls. PPL indicates perplexity.**

| Dataset | Query | $\log(\text{PPL})(\downarrow)$ | Rouge-L($\uparrow$) |
|---|---|---|---|
| Pereira's | w/o Brain | 2.219* | 0.213* |
| | RS Brain | 1.967* | 0.267* |
| | Brain-Aug | **1.946** | **0.272** |
| Huth's | w/o Brain | 3.573* | 0.148* |
| | RS Brain | 3.111* | 0.159* |
| | Brain-Aug | **2.997** | **0.167** |
| Narratives | w/o Brain | 4.328* | 0.083* |
| | RS Brain | 3.532* | 0.105* |
| | Brain-Aug | **3.471** | **0.109** |

(1) Regardless of whether BM25 or RepLLaMa is used as the ranking model, Brain-Aug substantially outperforms the original query and Unsup-Aug. According to NDCG@20 results, Brain-Aug improved the original query by 0.027 on Pereira's dataset, 0.014 on Huth's dataset, and 0.024 on the Narratives dataset. The only exception is observed when using RepLLaMa and metric MAP on Pereira's dataset. A possible explanation for this exception is the RepLLaMA's high performance on the Pereira dataset, which we discuss in observation (3).

(2) When considering various datasets and metrics, the Unsup-Aug query does not consistently outperform the original query. Significant differences between the performance achieved by the Unsup-Aug query and the original query emerge on the metric of Recall@20 when using BM25 as the ranking model. This observation suggests that Unsup-Aug, which improves query representation by tackling term mismatch issues, leads to an improvement in recall. When Brain-Aug is combined with Unsup-Aug (Unsup+Brain-Aug), we observe a performance gain when compared to Unsup-Aug. This highlights the effectiveness of brain signals in query augmentation and underscores the potential of combining them with traditional signals.

(3) We observe little difference in performance between RepLLaMa and BM25 on Huth's dataset and Narratives's dataset. This implies that in a zero-shot setting and cross-domain scenario (the datasets are derived from spoken stories, which differs from the training data of RepLLaMa), dense retrieval models like RepLLaMa are not necessarily better than traditional sparse retrieval models like BM25. This phenomenon is also observed in the BEIR dataset [51]. However, in Pereira's dataset, RepLLaMa shows significant improvement over BM25 with different query inputs. The impressive performance of RepLLaMa on Pereira's dataset can likely be attributed to the fact that the data in Pereira are likely to be used in the original construction of RepLLaMa.

#### 5.2.2 *Decomposing Brain-Aug.* Next, we investigate the contribution of brain signals and the ranking-oriented inference approach to Brain-Aug. Experimental results are presented in Table 3. First, we observe that removing (w/o Brain) or random sampling the brain inputs (RS Brain) leads to a decrease in performance. This indicates

**Table 2: Document ranking performance averaged across participants, with our method (*Brain-Aug* & *Brain+Unsup*) marked by a ⋆. Best results are in boldface, and the second-best results are underlined. ∗/† indicates Brain-Aug / Brain+Unsup significantly outperforms the baseline ($p \leq 0.05$, paired t-test), respectively.**

| Dataset | Query | BM25 | | | | RepLLaMA | | | |
|---|---|---|---|---|---|---|---|---|---|
| | | N@10 | N@20 | R@20 | MAP | N@10 | N@20 | R@20 | MAP |
| Pereira's | original | 0.643*,† | 0.664*,† | 0.888*,† | 0.594*,† | 0.878 | 0.881*,† | 0.964*,† | 0.858 |
| | Unsup-Aug | 0.646*,† | 0.655*,† | 0.924*,† | 0.590*,† | 0.872*,† | 0.877*,† | 0.951*,† | 0.855 |
| | Brain-Aug⋆ | 0.671 | **0.691** | **0.941** | **0.618** | **0.883** | **0.887** | **0.980** | **0.859** |
| | Unsup+Brain-Aug⋆ | **0.673** | 0.686 | 0.936 | 0.615 | 0.878 | 0.882 | 0.975 | 0.853 |
| Huth's | original | 0.297*,† | 0.326*,† | 0.536*,† | 0.264*,† | 0.299*,† | 0.328*,† | 0.520*,† | 0.275*,† |
| | Unsup-Aug | 0.291*,† | 0.320*,† | 0.575† | 0.259*,† | 0.302*,† | 0.333*,† | 0.537*,† | 0.276*,† |
| | Brain-Aug⋆ | 0.306 | 0.340 | 0.569† | **0.273** | **0.310** | **0.342** | 0.550 | **0.281** |
| | Unsup+Brain-Aug⋆ | **0.309** | **0.342** | **0.580** | 0.269 | 0.308 | 0.340 | **0.552** | 0.279 |
| Narratives | original | 0.419*,† | 0.434*,† | 0.629*,† | 0.355*,† | 0.413*,† | 0.426*,† | 0.611*,† | 0.351*,† |
| | Unsup-Aug | 0.440 | 0.452† | 0.670† | 0.367*,† | 0.416*,† | 0.431*,† | 0.629*,† | 0.356*,† |
| | Brain-Aug⋆ | 0.441 | 0.458 | 0.669 | **0.382** | 0.430 | **0.446** | 0.641 | **0.382** |
| | Unsup+Brain-Aug⋆ | **0.445** | **0.462** | **0.678** | **0.382** | **0.432** | **0.446** | **0.642** | 0.380 |

**Table 3: Document ranking performance of *Brain-Aug (ours)* and its controls with ranking model BM25. Best results in boldface. * indicates $p \leq 0.05$ for the paired t-test of *Brain-Aug* and the baseline.**

| Dataset | Query | NDCG@20 | MAP |
|---|---|---|---|
| Pereira's | w/o Brain | 0.665* | 0.586* |
| | RS Brain | 0.678* | 0.604* |
| | w/o IDF | 0.684* | 0.609* |
| | Brain-Aug | **0.691** | **0.618** |
| Huth's | w/o Brain | 0.332* | 0.265* |
| | RS Brain | 0.321* | 0.256* |
| | w/o IDF | 0.332* | 0.266* |
| | Brain-Aug | **0.340** | **0.273** |
| Narratives | w/o Brain | 0.452* | 0.368* |
| | RS Brain | 0.448* | 0.367* |
| | w/o IDF | 0.450* | 0.373* |
| | Brain-Aug | **0.458** | **0.382** |

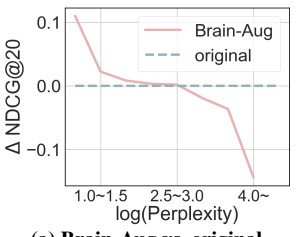 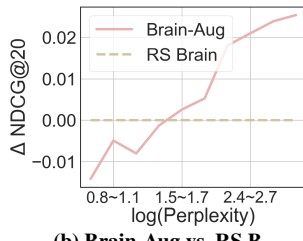

(a) Brain-Aug vs. original    (b) Brain-Aug vs. RS B

**Figure 2: Relationship between document ranking performance and perplexity of ground-truth query continuation in Pereira's dataset. "RS B" indicates the ablation of Brain-Aug that randomizes brain inputs. Δ NDCG@20 indicates performance gains of Brain-Aug.**

that semantic information decoded from brain signals within the query context enhances the query. Furthermore, while RS Brain consistently outperforms w/o Brain approach in terms of generation perplexity (see Section 5.1), it struggles to achieve better document ranking performance on the Huth's and Narratives datasets. This can be attributed to the fact that RS Brain, despite generating content that closely matches the token distribution of the whole dataset and reducing perplexity, fails to effectively differentiate between different documents within the dataset without semantics related to the query context. Last, we also observe a significant performance improvement when comparing Brain-Aug against its ablation without ranking-orient generation (w/o IDF). This suggests the importance of generating content that can be used to differentiate between documents.

*5.2.3 Relationship between document ranking and query generation performance.* Fig. 2 illustrates the relationship between the document ranking performance of Brain-Aug and RS Brain and the perplexity of query continuation measured using RS Brain. The lower perplexity of query generation indicates a higher likelihood of generating more accurate query continuation. This higher likelihood, as shown in Fig. 2a, further leads to an increase in document ranking performance. Conversely, Fig. 2b shows a different trend: when the perplexity is higher, the performance gain of Brain-Aug with its ablation RS Brain is higher. This implies that when generating accurate query continuations is difficult, semantics decoded from the query context with brain signals is more beneficial. This observation is consistent with findings by Ye et al. [54] that the addition of brain signals lead to a more substantial performance improvement when generating continuations with higher uncertainty.

*5.2.4 Example cases.* Table 4 presents example cases with the original query "The shaking can" which is sampled from document $d_{13}$ in Pereira's dataset. Brain-Aug leverages brain signals to expand

**Table 4: Examples of document ranking with BM25 using the original query or the augmented query in Pereira's dataset. Text in blue and in purple indicates content in the original query and generated by the query augmentation method, respectively.**

| Method | Query Content | Top-ranked document | Relevance |
|---|---|---|---|
| Original | The shaking can | $d_{21}$: The wind from the hurricane shook the house, shattering a window ... Later that night, with the wind shaking the house, ... | 0 |
| Unsup-Aug | The shaking can from house wind | $d_{21}$: The wind from the hurricane shook the house, shattering a window ... Later that night, with the wind shaking the house ... | 0 |
| RS Brain | The shaking can last anywhere from a few seconds to several minutes | $d_{21}$: The wind from the hurricane shook the house, shattering a window in the kitchen. ... Later that night, with the wind shaking the house, we fell asleep huddled on the sofa. | 0 |
| Brain-Aug | The shaking can be caused by an earthquake | $d_{13}$: Earthquakes shake the ground and can knock down buildings and other structures. also trigger landslides and volcanic activity. Most earthquakes are caused by ... | 1 |

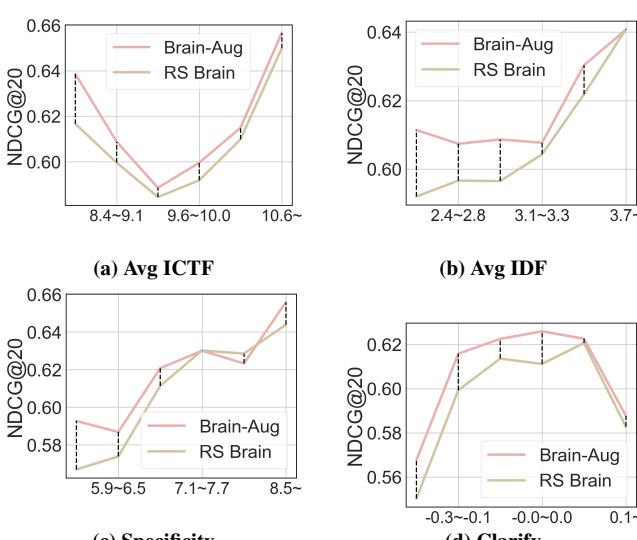

(a) Avg ICTF  (b) Avg IDF
(c) Specificity  (d) Clarify

**Figure 3: Document ranking performance w.r.t. different query features in Pereira's dataset.**

the query with "be caused by an earthquake". As a result, the relevant document with the topic of the earthquake, $d_{13}$, is appropriately ranked at the top of the search results. In contrast, when using the original query or augmenting it with unsupervised signals or randomly sampled brain signals, the document $d_{21}$, which discusses shaking wind, is erroneously ranked as the top result. This case study demonstrates the significant impact of incorporating brain signals into the query augmentation process. Example cases for Huth's and Narratives dataset are provided in Section A.6.

**Answer to RQ2.** We verified that a query augmented with semantics decoded from brain signals can significantly enhance document ranking performance. This performance enhancement is more pronounced when the generated query continuation is more accurately aligned with the query context.

## 5.3 Query performance analysis

Next, we investigate the performance improvement achieved by Brain-Aug for different queries by grouping queries according to their features. We select four query features: three pre-retrieval features (calculated based on query tokens), i.e., *ICTF*, *IDF*, and *specificity* score [49], and one post-retrieval feature (calculated based on the information of retrieved documents), i.e., *clarify* score [11, 35]. For details on the query features, see Section A.3. We conjecture that larger feature values correspond to a more clarified query and usually result in better retrieval quality.

Fig. 3 depicts the document ranking performance w.r.t. different query features on Pereira's dataset. We have two key observations. (i) When the averaged IDF, specificity score, and clarity score increase, both Brain-Aug and the RS Brain show an improvement in retrieval performance. This indicates that a more specific query usually has a better retrieval performance. (ii) The performance gain of Brain-Aug compared to RS Brain is more pronounced when these features experience a decrease. This observation is supported by a significant negative Pearson's $r$ between the improvement in NDCG@20 for Brain-Aug compared to RS Brain and the averaged ICTF, averaged IDF, specificity score, and clarity score, which are $-0.14$, $-0.19$, $-0.17$, and $-0.32$, respectively. This indicates that the performance improvement brought by brain signals is larger in queries prone to be vague or ambiguous.

**Answer to RQ3.** We have observed that queries prone to ambiguity (e.g., containing tokens with lower IDF scores or with low clarify scores) stand to gain more from Brain-Aug.

## 6 DISCUSSION AND CONCLUSION

### 6.1 Summary of Contributions

Existing research incorporating physiological signals in IR tasks, whether based on eye-tracking [6] or brain signals [14, 55], has relied on predicting relevance of presented information. Here, we have investigated an alternative approach for directly augmenting queries based on the semantic information decoded from fMRI brain signals. Our findings revealed that decoding semantic representations from brain signals can enhance the generation of queries and subsequently improving document ranking. Moreover, we have observed that brain signals are more effective when the content to be generated has

higher perplexity, indicating that decoded semantic information for unlikely query augmentations is more effective than it is for likely query augmentations. In conclusion, our findings open a horizon for new types of methods for understanding users by decoding semantics associated with information needs directly from brain signals. This process can kick off naturally as it happens as part of perceiving information and without requiring users to engage with any particular interaction technique or user interface.

## 6.2 Limitations and Future work

Our work has the following limitations pointing towards promising avenues for future research: (i) Our study utilized fMRI signals, which are not readily accessible in real-world human-computer interaction scenarios and have a significant delay of 2-8 seconds. More commonly used signals, such as electroencephalogram (EEG), have lower signal-to-noise ratios, which may limit their utility for semantic decoding. Currently, there is a lack of evidence that EEG can effectively decode semantics. The significant component of Brain-Aug is designed to be independent of the type of signal employed. This paper chose to use fMRI signals because fMRI has been extensively studied in semantic decoding among various physiological signals (e.g., EEG, MEG, ECoG, and fNIRS). In recent years, sensor technology like Functional near Infrared Spectroscopy (fNIRS) and MEG may become promising directions for future research. With the advancement in the quality of brain data collected by these lightweight devices, we believe that Brain-Aug has the potential to be applied in more realistic scenarios, including IR, virtual reality, disabled services, and intelligent assistants. (ii) Our experiments simulate the the ranking task on fMRI datasets following Izacard et al. [18] and Lee et al. [25]. The simulation shows significant improvements in Brain-Aug over the baselines and carefully designed controls. Although the simulation methodology is commonly used to test retrieval performance, it is different from realistic user interactions. This simulation was driven by its advantage in building a sufficient number of queries and obtaining the corresponding query context to construct a substantial amount of training data. In the future, we plan to explore evaluations that closely resemble real-world tasks for search engine users. Despite the high cost of conducting these experiments, we believe it can help advance this promising field.

## 6.3 Ethical considerations

Recently, there has been a series of works attempting to utilize brain–computer interface (BCI) technology to enhance information accessing performance in various language-related applications, such as search [3, 13, 44] and communication [42]. Such technology is currently at a very early stage where such applications feel a long way off. However, it is important to discuss the associated concerns regarding privacy issues as the collection of brain signals is inherently susceptible to the actions of malicious third parties, which increases the risk of potential misuse or mishandling of sensitive information.

On the one hand, raw data collected via neurophysiological devices should be treated as private information, as such data can potentially be used to identify an individual [4] as well as their physiological disorders and thoughts [59]. This technology may lead to

risks such as influencing people's political opinions, and discrimination during recruiting based on their neural profiles. Therefore, the raw data should be avoided from being uploaded to the cloud for computation. It is necessary to filter sensitive information and decode only the information that helps the user accomplish their task with local computing. For publicly available datasets, ethical review and informed consent from each participant should be obtained, such as the dataset used in this paper (see Section A.1). Additionally, datasets should be used strictly for research purposes following their respective licenses.

On the other hand, there is a concern regarding the interaction log that might be recorded in applications like search engines. Although such interactions, such as clicks, comments, and submitted queries, are frequently recorded for improving individual user experience, the utilization of BCI can potentially pose greater risks. For example, it can be employed to capture users' genuine opinions on content within information systems, which can then be adopted in applications such as selective exposure and targeted advertising. Hence, users should have the right to decide whether they are willing to provide their interaction history to service providers. This is already specified in the legislation of many countries. In addition, the interaction history, even with users' permission, should undergo post-hoc filtering to remove any sensitive information before being utilized to train a model aimed at enhancing the commercial product.

## 7 REPRODUCIBILITY

Our experiments use open-source datasets (Pereira's dataset [42], Huth's dataset [24], and the Narratives dataset [41]), which can be downloaded from the paper websites or OpenNeuro[1]). The data from Pereira et al. [42] is available under the CC BY 4.0 license. The Huth's dataset and Narratives dataset are provided with a "CC0" license. Code is released using an anonymous link during the review process: https://anonymous.4open.science/r/Brain-Query-Augmentation-B6CC/. All codes used in the paper are available under the MIT license after the review process.

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
