# OpenReview forum: "Query Augmentation with Brain Signals"
_acmmm.org/ACMMM/2024/Conference — MM2024 Oral_

### Official Review · Reviewer_UVw1 · 2024-05-22

**Rating:** 5
**Confidence:** 3

**Summary:**

The paper "Query Augmentation with Brain Signals" introduces Brain-Aug, a novel approach for enhancing query representation by decoding semantic information directly from brain signals. Brain-Aug employs a threefold technique involving an adapter network to map brain signals into a language model's embedding space, a training protocol using prompt tuning, and a ranking-oriented inference strategy. The method significantly improves document ranking accuracy, especially for ambiguous queries, using multiple fMRI datasets.

**Strengths:**

- **Novelty**: Brain-Aug leverages brain signals for query augmentation, a unique approach that differentiates it from traditional methods relying on user interactions or external document content.
- **Technical Correctness**: The methodology integrates brain signals with language models through a well-designed neural network and prompt tuning, ensuring accurate semantic decoding.
- **Evaluation**: The use of multiple fMRI datasets to validate the effectiveness of Brain-Aug demonstrates thorough and diverse testing, enhancing the reliability of the results.
- **Applications**: This approach has significant potential for improving information retrieval systems by better understanding and aligning with user intentions, especially for complex or ambiguous queries.

**Limitations:**

- **Accessibility**: The reliance on fMRI signals limits the practical applicability of the method, as fMRI is not readily accessible in everyday scenarios.
- **Latency**: fMRI signals have a significant delay (2-8 seconds), which may affect real-time applications.
- **Simulation vs. Real-world Testing**: The experiments were conducted in simulated environments using fMRI datasets, which may not fully represent real-world user interactions and query contexts.
- **Generality**: The method’s effectiveness is contingent on the quality and specificity of the fMRI data, which may not be consistent across different users or contexts.

**Suitability:**

2

---

### Official Review · Reviewer_8NKq · 2024-05-24

**Rating:** 3
**Confidence:** 3

**Summary:**

This paper introduces a new perspective on enhanced retrieval of queries with semantic information decoded from fMRI. Experiments demonstrate that Brain-Aug enhances query performance.

**Strengths:**

* The motivation of this paper is interesting, solid, and potentially valuable for future research.
* The paper is clearly written and easy to read.
* Extensive experiments are conducted to analyze the effectiveness of the proposed method.

**Limitations:**

**Implementation(inappropriate):**

* The method proposed in this paper is based on the enhancement of human brain activity during active query. However, in the dataset used such as Pereira’s dataset [1], the fMRI signals were all acquired based on subjects receiving semantic stimuli. There is a fundamental difference between the two and the authors do not explain the reasonableness for using this type of dataset.

**Method (lack of novelty):**

* The novelty of this paper is limited. Decoding semantic information from fMRI has been proposed in [2, 3], prompt-tuning for large language models has been studied in [4]. So the authors should have emphasised their adaptive design in this task, rather than a straightforward combination of existing techniques.

**Experiment (unfair comparison):**

* The authors emphasise the advantages of Brain-Aug over previous retrieval enhancement methods, but fail to experimentally compare it with any related methods, and the overall experimental presentation is more like an ablation experiment. If the dataset does not have what is needed, such as historical clicked documents, then a simple construction similar to Brain-Aug is necessary.

* The results of RS Brain and Brain-Aug are so close to each other in Table 1 that it is still doubtful whether the improvement brought by Brain-Aug is simply due to the provision of longer textual information. This can be resolved when Brain-Aug's predicted tokens are replaced with text generated by fMRI decoding models for comparison, which are available in datasets such as the Pereira’s dataset [1].

**Reference**

[1] Francisco Pereira, Bin Lou, Brianna Pritchett, Samuel Ritter, Samuel J Gershman, Nancy Kanwisher, Matthew Botvinick, Evelina Fedorenko: Toward a universal decoder of linguistic meaning from brain activation. Nature Communications 2018

[2] Jerry Tang, Amanda LeBel, Shailee Jain, Alexander G. Huth: Semantic reconstruction of continuous language from non-invasive brain recordings. Nature Neuroscience, 2023

[3] Yiqun Duan, Jinzhao Zhou, Zhen Wang, Yu-Kai Wang, Chin-Teng Lin:DeWave: Discrete EEG Waves Encoding for Brain Dynamics to Text Translation. NeurIPS 2023

[4] Mustafa Özdayi, Charith Peris, Jack FitzGerald, Christophe Dupuy, Jimit Majmudar, Haidar Khan, Rahil Parikh, Rahul Gupta:
Controlling the Extraction of Memorized Data from Large Language Models via Prompt-Tuning. ACL 2023

**Suitability:**

3

---

### Official Review · Reviewer_RAbA · 2024-05-24

**Rating:** 4
**Confidence:** 2

**Summary:**

This paper utilizes fMRI signals of brain activity to enhance information retrieval. To achieve this, authors use prompt tuning to train the fMRI encoder and jointly input fMRI embeddings and query embeddings into trained LLM to generate better queries.

**Strengths:**

+ This paper focuses on a very interesting research question.
+ The authors conduct extensive evaluations of proposed methodology on three datasets.

**Limitations:**

**Regarding the reason of performance improvement:**

According to Table 1, I find that the improvement of “RS brain” over “w/o brain” is much higher than that of “Brain-Aug” over “RS brain”. This seems to indicate that the main reasons of query generation performance improvement are the added learnable prompts (<b> and <\b>) and conducting fine-tuning, while the introduction of fMRI seems to be of secondary importance.

**Regarding evaluation:**

According to line 337-348, the evaluation task are different from those described in the introduction section. This difference raises my concern as to whether the method is valid for the task described in the introduction.



I consider to revise my rating based on the authors' rebuttal.

**Suitability:**

3

---

### Meta-Review · Area_Chair_Xkt7 · 2024-06-29

**Recommendation:** Accept (Oral)
**Confidence:** 4

**Metareview:**

This paper presents Brain-Aug, a new method that enhances query representation by translating brain signals into a language model's embedding space. Using an adapter network, prompt tuning, and a ranking-focused inference strategy, Brain-Aug significantly boosts document ranking accuracy for ambiguous queries across various fMRI datasets.

While the author's rebuttal addressed most of my concerns, there remain a few outstanding issues. Specifically, the evaluation task's deviation from the initially envisioned scenario is still problematic and requires further clarification. Additionally, the exploration of the implications of BrainLLM, including its potential for active activity or stimulation, needs to be thoroughly discussed to enhance the paper's credibility. Addressing these aspects comprehensively in the final version would significantly strengthen the paper.